# Multiplayer Information Asymmetric Bandits in Metric Spaces

## Abstract

In recent years the information asymmetric Lipschitz bandits In this paper we studied the Lipschitz bandit problem applied to the multiplayer information asymmetric problem studied in Chang et al. (2022); Chang and Lu (2023). More specifically we consider information asymmetry in rewards, actions, or both. We adopt the CAB algorithm given in Kleinberg (2004b) which uses a fixed discretization to give regret bounds of the same order (in the dimension of the action) space in all 3 problem settings. We also adopt their zooming algorithm Kleinberg et al. (2008)which uses an adaptive discretization and apply it to information asymmetry in rewards and information asymmetry in actions.

## 1 Introduction

The multi-armed bandit problem is a classical problem in reinforcement learning, where an agent is presented with a finite set of actions (or arms) to choose from. Selecting an arm yields a stochastic reward that is assumed to be 1-subgaussian and bounded in the interval $[0, 1]$. Since the rewards are bounded, so are their expectations; thus, we denote the true mean reward of each arm $a$ as $\mu_a \in [0, 1]$. The goal is to maximize the cumulative expected reward over a finite time horizon by selecting arms with high $\mu_a$ as frequently as possible.

There has been growing interest in a variant of this setting where the action space is no longer finite. This is known as the Lipschitz bandit setting, in which actions are indexed by a compact subset of a metric or Euclidean space. In such settings, algorithms designed for finite arm sets—such as the UCB algorithm of Lai and Robbins (1985)—are no longer directly applicable, since UCB requires every arm to be pulled at least once, which is infeasible when the set of arms is uncountably infinite.

To address this, the Lipschitz bandit setting introduces a structural assumption: the Lipschitz condition. Specifically, for any two actions $a$ and $b$, the absolute difference in their true means satisfies

$$|\mu_a - \mu_b| \leq L \cdot |a - b|, \tag{1}$$

where $L$ is a known Lipschitz constant and $|\cdot|$ denotes a norm on the action space. Intuitively, this condition implies that if an agent has a good estimate of the mean reward at one action, then nearby actions must have similar rewards. This structural smoothness allows generalization from observed rewards to unobserved actions.

In the single-player case, the Lipschitz bandit problem was first studied by Kleinberg (2004b), who proposed a fixed discretization approach. They partition the action space into a finite grid (the `CAB` algorithm) and apply the classical UCB algorithm on this discretized set. Later, Kleinberg et al. (2008) introduced a more adaptive approach known as the zooming algorithm, which dynamically refines the discretization in regions where the rewards appear promising. This algorithm achieves a regret bound of order $O(T^{\frac{d+1}{d+2}})$, where $d$ is the covering dimension of the action space, and also matches this with a corresponding lower bound.

More recently, there has been increasing attention on multiplayer bandit settings, with applications in distributed control and wireless networks. Many of these works assume a communication graph between players or consider settings without joint actions. In contrast, our focus is on the information-asymmetric multiplayer bandit model introduced by Chang et al. (2022); Chang and Lu (2023). In this setting, each

of the $n$ players has access to a private set of actions, and in each round, all players simultaneously select one action from their own set, resulting in a joint action. The overall action space is thus exponential in the number of players.

We are particularly interested in three types of information asymmetry in the multiplayer setting. The first, which we refer to as Problem A, involves asymmetry in actions: all players receive the same reward, determined by the joint action, but each player is unaware of the actions chosen by the others. The second, Problem B, concerns asymmetry in rewards: players can observe the full joint action but receive independent and identically distributed (i.i.d.) rewards, meaning the feedback is private to each player even though the action profile is shared. The third and most challenging setting, Problem C, involves asymmetry in both actions and rewards: players neither observe the actions of others nor share a common reward signal. Instead, each player receives their own i.i.d. reward based solely on the joint action, with no visibility into the behavior or feedback of others.

**Our contribution** In this paper, we propose the first algorithmic framework for the multiplayer information-asymmetric Lipschitz bandit setting. We formalize three distinct problem variants—Problems A, B, and C—corresponding to asymmetry in actions, rewards, and both, respectively. To address these challenges, we leverage a combination of uniform discretization and recent multi-agent algorithms introduced in Chang et al. (2022); Chang and Lu (2023) to handle the information asymmetries inherent in all three problems. Furthermore, we adapt the zooming algorithm to accommodate settings with asymmetry in actions (Problem A) and in rewards (Problem B). For these two cases, we are able to prove regret bounds that match the best-known rates in the single-player setting. For Problem C, which involves both forms of asymmetry, we provide an algorithm with a regret bound that is nearly optimal up to logarithmic factors.

## 1.1 Related Works

**Lipschitz Bandits.** There is a rich body of work on Lipschitz bandit problems. One major line of research considers the case where the Lipschitz constant is unknown. One of the earliest works in this direction is Jones et al. (1993), which studies global optimization under Lipschitz continuity without access to the constant. They modify Shubert's algorithm to reduce computational cost and mitigate excessive global exploration. Bubeck et al. (2011b) address the stochastic Lipschitz bandit problem with unknown Lipschitz values by estimating the constant and optimizing for worst-case regret bounds. Similarly, Lu et al. (2010) study Lipschitz contextual bandits over a metric space, proposing the query-ad-clustering algorithm that partitions the space into "good" and "bad" regions and uses lower bounds to guide exploration.

A significant extension appears in Kleinberg et al. (2019), which generalizes Lipschitz bandits to arbitrary metric spaces, going beyond the classical $[0, 1]$ setting with $\ell_1^{1/d}$ metrics. They ask which metric spaces admit sublinear regret and what the optimal regret rates are in these spaces. Bubeck et al. (2011a) propose a model with invariant structure, where arms and rewards are symmetric under known transformations. They introduce the UniformMesh-N algorithm, which leverages group orbits to incorporate side observations into a uniform discretization framework. In the discrete Lipschitz setting, Magureanu et al. (2014) propose OSLB, an algorithm matching lower bounds on regret, and CKL-UCB, for which they also provide a finite-time analysis. Lu et al. (2019) tackle the problem with heavy-tailed reward distributions, introducing the SDTM algorithm, which combines standard discretization with truncated means due to the failure of UCB in such settings.

Other extensions include Krishnamurthy et al. (2020), who study contextual Lipschitz bandits with smoothed regret guarantees under minimal continuity assumptions. They propose algorithms that learn mappings from contexts to distributions over arms. Wang et al. (2020b) relate tree-based methods to Gaussian processes in the Lipschitz setting, introducing TreeUCB, which maintains a UCB index for each region and adaptively fits a decision tree to past observations. Feng et al. (2022) study a batched setting, where rewards are observed in groups, and propose BLiN, a batched zooming algorithm achieving nearly optimal regret.

**Multiplayer Bandits.** A parallel line of research investigates multiplayer bandit problems. In cooperative bandits, the players aim to identify the best arm from a shared action set. Communication is typically modeled

via a graph, where edges represent the ability to exchange information. This framework was introduced in Awerbuch and Kleinberg (2008), followed by strategies such as $\epsilon$-greedy Szorenyi et al. (2013), gossip-based UCB Landgren et al. (2016), and its accelerated variant Martínez-Rubio et al. (2019). The "adopting a leader" strategy is studied in Wang et al. (2020a). The adversarial version of this problem is considered in Bar-On and Mansour (2019), where followers use an EXP3-style algorithm. Additional work has explored delayed feedback in neighbor-sharing settings Cesa-Bianchi et al. (2016), and asynchronous participation, where only a subset of players is active at any given time Bonnefoi et al. (2017); Cesa-Bianchi et al. (2020).

In the collision setting, players choose arms independently, and if multiple players select the same arm, a collision occurs and no rewards are obtained. Unlike our setting, joint actions are not considered. Proutiere and Wang (2019) extend this to the Lipschitz case, proposing the DPE (Decentralized Parsimonious Exploration) algorithm, which minimizes communication and achieves near-optimal regret.

Another model is competing bandits, introduced in Liu et al. (2020), where players have preferences, and when multiple players select the same arm, only the top-ranked one receives the reward. A centralized version, CUB, is proposed where players send their UCB indices to a central agent. Cen and Shah (2022) show that with the ability to assign transfers between players and arms, logarithmic regret is achievable. Jagadeesan et al. (2021) extend this by allowing agents to negotiate these transfers under equilibrium constraints. Liu et al. (2020) also propose an ETC-style algorithm that achieves logarithmic regret when the reward gaps are known, and Sankararaman et al. (2021) eliminate this requirement. Finally, Liu et al. (2021) propose a decentralized UCB algorithm with built-in collision avoidance mechanisms.

## 2 Preliminary

We study the $L$-Lipschitz multiplayer bandit problem, where the joint action space is given by $[0,1]^{md}$. This space is decomposed as the direct product of $m$ copies of $[0,1]^d$, where each $[0,1]^d$ represents the action space of an individual player. At each round $t$, all $m$ players simultaneously and independently select an arm from their respective spaces, without any communication. The resulting joint action $\boldsymbol{a}_t = (a_t^1, \ldots, a_t^m)$ lies in $[0,1]^{md}$ and is formed by the Cartesian product of the players' individual choices.

We assume the reward function $\mu : [0,1]^{md} \to [0,1]$ satisfies the following Lipschitz condition: for any two joint actions $\boldsymbol{a}, \boldsymbol{a}' \in [0,1]^{md}$,

$$|\mu_{\boldsymbol{a}} - \mu_{\boldsymbol{a}'}| \leq L|\boldsymbol{a} - \boldsymbol{a}'|, \tag{2}$$

where $|\cdot|$ is any norm on $\mathbb{R}^{md}$ and $L > 0$ is a known Lipschitz constant. For convenience, we denote by $B(\boldsymbol{a}, r)$ the closed ball of radius $r$ centered at joint arm $\boldsymbol{a}$.

Let $\mu^* = \sup_{\boldsymbol{a} \in [0,1]^{md}} \mu_{\boldsymbol{a}}$ denote the expected reward of the optimal joint arm. At round $t$, let $\boldsymbol{a}t$ be the joint arm played, and let $X\boldsymbol{a}t$ be the stochastic reward received. The expected regret after $T$ rounds is defined as:

$$R_T = T\mu^* - \sum t = 1^T \mathbb{E}[X_{\boldsymbol{a}t}] = \sum t = 1^T \Delta_{\boldsymbol{a}t}, \tag{3}$$

where $\Delta\boldsymbol{a} := \mu^* - \mu_{\boldsymbol{a}}$ denotes the suboptimality gap of joint arm $\boldsymbol{a}$.

In this paper, we study the $L$-Lipschitz multiplayer bandit problem under various forms of information asymmetry, as introduced in earlier works such as Chang et al. (2022); Chang and Lu (2023). In all cases, players are permitted to agree on a joint strategy before the learning process begins, but once the interaction starts, no further communication or coordination is allowed.

**Problem A: Asymmetry in Actions.** In this setting, each player receives the same stochastic reward $X_{\boldsymbol{a}_t}$ based on the joint action, but is unable to observe the actions chosen by the other players. The challenge lies in coordinating implicitly through shared feedback without knowledge of the others' behavior.

**Problem B: Asymmetry in Rewards.** Here, players are able to observe the full joint action $\boldsymbol{a}_t$, i.e., the individual actions of all players. However, each player receives their own independent and identically distributed (i.i.d.) copy of the reward based on $\boldsymbol{a}_t$. That is, they receive separate realizations of a shared underlying stochastic process and cannot observe each other's rewards.

**Problem C: Asymmetry in Both Actions and Rewards.** This is the most restrictive setting. Each player is unable to observe the actions of the others and receives their own i.i.d. realization of the reward. Thus, players lack both action-level and reward-level visibility, making coordination and learning significantly more difficult.

It is important to note that in both Problems B and C, although each player receives a separate reward realization, these are i.i.d. samples from the same distribution. Since we measure regret in expectation, all players incur the same regret in these settings. The central difficulty in Problems B and C arises not from differences in the experienced regret, but from the lack of shared information, which limits the players' ability to adapt or coordinate during learning.

## 3 Uniform Discretization

### 3.1 From Lipschitz Regret to Discrete Bandit Regret

To apply MAB algorithms in the continuous Lipschitz setting, we discretize the action space. Each player partitions their individual domain $[0,1]$ into $K$ equal intervals by placing markers at $\left\{0, \frac{1}{K}, \frac{2}{K}, \dots, \frac{K-1}{K}\right\}$. The full discretized action space for each player, originally $[0,1]^d$, becomes the Cartesian product:

$$\left\{0, \frac{1}{K}, \frac{2}{K}, \dots, \frac{K-1}{K}\right\}^d.$$

Consequently, the joint discretized action space for all $m$ players is a subset of $[0,1]^{md}$ containing $K^{md}$ arms.

Let $\boldsymbol{a}_t \in [0,1]^{md}$ denote the joint continuous action taken at round $t$, and let $\mu^* = \sup_{\boldsymbol{a} \in [0,1]^{md}} \mu_{\boldsymbol{a}}$ be the optimal reward over the continuous domain. The regret in the continuous Lipschitz setting is given by:

$$R_T = T\mu^* - \sum_{t=1}^{T} \mu_{\boldsymbol{a}_t}.$$

We now introduce the regret for the discretized version of the problem. Let $\boldsymbol{a}_t \in \mathcal{A}_K \subset [0,1]^{md}$ denote the joint discrete action taken at round $t$, where $\mathcal{A}_K$ contains all joint actions formed from the $K$-way discretization. Let $\mu_K^* = \max_{\boldsymbol{a} \in \mathcal{A}_K} \mu_{\boldsymbol{a}}$ be the optimal reward over this finite set. Then the discrete regret of a policy $\pi$ is defined as:

$$R_K^\pi(T) = T\mu_K^* - \sum_{t=1}^{T} \mu_{\boldsymbol{a}_t}.$$

Note that since $\mathcal{A}_K \subset [0,1]^{md}$, it follows that $\mu_K^* \leq \mu^*$.

We now relate the regret in the continuous setting to that in the discretized setting via the Lipschitz condition:

**Lemma 1** *Suppose the regret of a policy $\pi$ on a discretized action space of $K^{md}$ arms over horizon $T$ is $R_K^\pi(T)$. Then the regret $R_T$ in the continuous Lipschitz bandit setting satisfies:*

$$R_T \leq \frac{L\sqrt{md}\,T}{K} + R_K^\pi(T),$$

*where $L$ is the Lipschitz constant and $\|\cdot\|$ denotes the $\ell_2$ norm on $\mathbb{R}^{md}$.*

This result shows that discretization introduces an additive error term of order $O(T/K)$ due to the coarseness of the discretized space, while enabling the application of standard MAB algorithms. By choosing $K$ appropriately as a function of $T$, we can balance this discretization error against the regret of the finite-arm policy to achieve optimal or near-optimal regret in the original continuous setting.

In each of these settings, it is useful to define, for each joint arm $\boldsymbol{a}$, a corresponding *upper confidence bound (UCB)* index. At time $t$, we define

$$\mathrm{UCB}_{\boldsymbol{a}}(t) := \widehat{\mu}_{\boldsymbol{a}}(t) + \underbrace{\sqrt{\frac{6 \log T}{n_{\boldsymbol{a}}(t)}}}_{:=\epsilon_{\boldsymbol{a}}(t)},$$

where $\widehat{\mu}_{\boldsymbol{a}}(t)$ is the empirical mean of arm $\boldsymbol{a}$ up to time $t$, and $n_{\boldsymbol{a}}(t)$ is the number of times arm $\boldsymbol{a}$ has been selected by time $t$. The term $\epsilon_{\boldsymbol{a}}(t)$ acts as a confidence radius or error term, which bounds the deviation between the empirical and true means. Notably, $\epsilon_{\boldsymbol{a}}(t)$ is independent of the specific reward distribution and depends only on the number of times arm $\boldsymbol{a}$ has been pulled and the time horizon $T$. As $n_{\boldsymbol{a}}(t)$ increases, $\epsilon_{\boldsymbol{a}}(t)$ decreases, allowing for increasingly accurate estimates.

We now present the proof of Lemma 1.

**Proof:** Recall the discretization procedure described earlier, and define $R_K^{\pi}(T)$ as the regret incurred by the policy $\pi$ over the discretized action set $\mathcal{A}_K$. Let $\boldsymbol{a}^*$ denote the optimal (continuous) arm achieving reward $\mu^*$. Let $\boldsymbol{a} \in \mathcal{A}_K$ be a discretized arm such that $\boldsymbol{a}^* \in \boldsymbol{a} + [0, \frac{1}{K}]^{md}$, i.e., $\boldsymbol{a}$ is the corner of the discretization cell (a hypercube) containing $\boldsymbol{a}^*$. Since our discretization uniformly tiles $[0, 1]^{md}$ into cubes of side length $1/K$, the maximum distance between $\boldsymbol{a}$ and $\boldsymbol{a}^*$ is at most $\frac{\sqrt{md}}{K}$.

We decompose the total regret as:

$$R_T = \underbrace{T\mu^* - T\mu_{\boldsymbol{a}}}_{(\mathrm{I})} + \underbrace{T\mu_{\boldsymbol{a}} - T\mu_K^*}_{(\mathrm{II})} + \underbrace{T\mu_K^* - \sum_{t=1}^{T} \mu_{\boldsymbol{a}_t}}_{(\mathrm{III})}.$$

We analyze each term:

- For term (I), since $\boldsymbol{a}$ and $\boldsymbol{a}^*$ are in the same cell, their distance is at most $\frac{\sqrt{md}}{K}$, and by the Lipschitz condition:
$$T\mu^* - T\mu_{\boldsymbol{a}} \le TL \|\boldsymbol{a}^* - \boldsymbol{a}\| \le \frac{L\sqrt{md}T}{K}.$$

- For term (II), since $\mu_K^* = \max_{\boldsymbol{b} \in \mathcal{A}_K} \mu_{\boldsymbol{b}} \ge \mu_{\boldsymbol{a}}$, we have:
$$T\mu_{\boldsymbol{a}} - T\mu_K^* \le 0.$$

- For term (III), this is precisely the discrete regret:
$$T\mu_K^* - \sum_{t=1}^{T} \mu_{\boldsymbol{a}_t} = R_K^{\pi}(T).$$

Putting these together, we obtain the total regret bound:

$$R_T \le \frac{L\sqrt{md}T}{K} + R_K^{\pi}(T),$$

as claimed. $\square$

### 3.2 Asymmetry in Arms (Problem A)

In this section, we study **Problem A**, the setting of *asymmetry in actions*. This models a multiplayer bandit scenario in which each player is unable to observe the actions of the other players. All players receive the same reward, which is determined by the joint action taken at each round.

In the standard single-player bandit setting, the player observes both the arm selected and the corresponding reward, which allows them to balance exploration and exploitation effectively. However, in the multiplayer setting, certain forms of information are hidden—namely, the other players' actions, their rewards, or both. The works Chang et al. (2022); Chang and Lu (2023) develop algorithms to handle these various forms of information asymmetry. A key challenge in such settings is the lack of communication: the players must coordinate without exchanging information during learning.

To apply algorithms developed for finite-armed bandits to continuous action spaces, prior work (e.g., Kleinberg (2004a)) uses a uniform discretization of the action space, reducing the problem to a finite one. The Lipschitz condition ensures that with sufficiently fine discretization, unsampled actions will not differ too much in reward from nearby sampled ones. In this paper, we adopt a similar discretization approach, but now in the context of *multi-agent* bandits with information asymmetry.

We propose **mCAB-A** (multi-agent Coordination Algorithm for Bandits — Asymmetry in actions), which extends finite-action algorithms to the asymmetric multiplayer setting. Each player discretizes their action space $[0,1]^d$ into the grid $\left\{0, \frac{1}{K}, \ldots, 1\right\}^d$, resulting in $K^d$ actions per player. The joint action space thus consists of $K^{md}$ total joint arms, where $m$ is the number of players.

The key idea of mCAB-A is that each player maintains a UCB index *for each joint arm*—not just the arms they themselves can pull. This poses a challenge: since each player cannot observe the other players' actions, it is unclear which joint arm was actually selected, and thus which UCB index should be updated.

To resolve this, the players agree *beforehand* on a deterministic ordering of all joint arms (e.g., lexicographic order). Since all players receive the same reward and follow the same algorithm, they remain synchronized in their estimates and UCB indices for each joint arm. As long as coordination is preserved, each player knows which arm was selected at each round—even though they cannot see the individual actions of the others.

At each round, every player computes the UCB index for all joint arms and selects the joint arm with the highest index. Ties are broken deterministically according to the predefined ordering. This ensures that the players remain coordinated and continue to select the same joint arms over time.

**Definition 2** *Number the players $1, \ldots, m$ and the $K$ individual actions, and consider each set of joint action $\boldsymbol{a}$ as an $m$ digit number with each digit corresponding to the joint action. Call this base $K$ number $N_{\boldsymbol{a}}$. For joint action $\boldsymbol{a}, \boldsymbol{b} \in \mathcal{A}$, we say that $\boldsymbol{a} < \boldsymbol{b}$ if $N_{\boldsymbol{a}} < N_{\boldsymbol{b}}$.*

Therefore, if all the players agree to pull the smallest arm as defined in the ordering above in the case of a tie, the players can infer the actions of the other players without having to observe them. The algorithm is stated more explicitly in Algorithm 9.

---

**Algorithm 1:** `mCAB-A` for asymmetry in actions

---

**1** **Input:** $T, M, d \in \mathbb{N}$

**2** Each player will discretize their actions space as $\{0, \frac{1}{K}, \frac{2}{K}, \ldots, 1\}^d$, where $K$ is given in equation equation 4

**3** Run `mUCB` on discretized joint action space Chang et al. (2022). More explicitly, **for** $t \leq K_{\max}$ **do**

**4**     Player $P_i$ will start from his arm 1 and successively pull each arm $K_{i+1} \cdots K_M$ times before moving to the next arm. They will repeat this entire epoch $K_1 \cdots K_{i-1}$ times.

**5** **end**

**6** **for** $t > K_{\max}$ **do**

**7**     Player $P_i$ chooses arm $a_i^*(t) = \arg\max_{a_i} \left( \max_{a_{-i}} \text{UCB}_{a_i, a_{-i}}^i(t) \right)$, which corresponds to player $i$ picking the $i$th component of $a^*(t)$ that maximises the index $\text{UCB}_a(t)$. In case of a tie between say $a$ and $a'$, they pick corresponding components of $a$ such that $a < a'$, where the order relation is as specified in Definition 2.

**8**     Player $P_i$ updates the UCB index $\text{UCB}_a^i(t+1)$ for arm $a$ setting $\delta = \frac{1}{T^2}$ with the received reward $X_{a^*(t)}^i(t)$.

**9** **end**

---

We can now present the regret bound for the Problem A algorithm.

**Theorem 3** *The regret of* `mCAB-A` *is given by*

$$R_T = O\left(T^{\frac{2Md+1}{2Md+2}} \cdot L^{\frac{Md}{Md+1}} \cdot (\log T)^{\frac{1}{2(Md+1)}}\right).$$

**Proof:** Suppose each player discretizes their action space $[0,1]^d$ into $K$ uniform bins along each coordinate. Then the total number of joint arms is $K^{Md}$. From Theorem 2 of Chang et al. (2022), we know that the regret $R_K(T)$ of `mUCB` on $K^{Md}$ arms is bounded as

$$R_K(T) = O\left(K^{Md} \cdot \sqrt{T \log T}\right).$$

By Lemma 1, the total regret in the continuous Lipschitz setting is bounded by

$$R_T \leq \frac{L\sqrt{Md} \cdot T}{K} + O\left(K^{Md} \cdot \sqrt{T \log T}\right).$$

To minimize this upper bound, we balance the two terms using the AM-GM inequality, or by solving for $K$ such that both terms are of the same order. That is,

$$\frac{L\sqrt{Md} \cdot T}{K} = K^{Md} \cdot \sqrt{T \log T}.$$

Solving this equation yields the optimal discretization scale:

$$K = \frac{T^{\frac{1}{2(Md+1)}} \cdot L^{\frac{1}{Md+1}}}{(\log T)^{\frac{1}{2(Md+1)}}}. \tag{4}$$

Substituting this optimal value of $K$ into the original regret bound yields

$$R_T = O\left(T^{\frac{2Md+1}{2Md+2}} \cdot L^{\frac{Md}{Md+1}} \cdot (\log T)^{\frac{1}{2(Md+1)}}\right).$$

$\square$

### 3.3 Asymmetry in Rewards (Problem B)

In this section, we propose an algorithm for **Problem B**, the setting of *asymmetry in rewards*. In this setting, each player observes the full joint action but receives an independent reward sample. This introduces a challenge not present in Problem A: since the rewards are i.i.d., different players receive different samples for the same joint arm, and thus maintain different empirical means and UCB indices. This lack of synchronization leads to inevitable *mis-coordination*, especially in the early rounds, as players' estimates diverge. See Chang et al. (2022) for a discussion of why the simple coordination scheme used in Problem A fails under this form of asymmetry.

However, we can leverage the observability of the joint actions to address this difficulty. The algorithm we propose is inspired by `mUCB-Intervals` from Chang and Lu (2023), adapted to the continuous-action Lipschitz setting via discretization. As in Problem A, we uniformly discretize the action space of each player. For any joint arm $\boldsymbol{a}$, with high probability we have:

$$\mu_{\boldsymbol{a}} \in I_{\boldsymbol{a}}(t) := \left(\widehat{\mu}_{\boldsymbol{a}}(t) - \underbrace{\sqrt{\frac{6\log T}{n_{\boldsymbol{a}}(t)}}}_{:=\epsilon_{\boldsymbol{a}}(t)}, \ \widehat{\mu}_{\boldsymbol{a}}(t) + \sqrt{\frac{6\log T}{n_{\boldsymbol{a}}(t)}}\right), \tag{5}$$

where $\widehat{\mu}_{\boldsymbol{a}}(t)$ is the empirical mean and $n_{\boldsymbol{a}}(t)$ is the number of times arm $\boldsymbol{a}$ has been pulled up to round $t$.

Under the "good event" that the above confidence intervals contain the true mean, we can eliminate arms as follows: if for two arms $\boldsymbol{a}$ and $\boldsymbol{b}$, the upper bound of $I_{\boldsymbol{a}}(t)$ is strictly less than the lower bound of $I_{\boldsymbol{b}}(t)$,

$$\widehat{\mu}_{\boldsymbol{a}}(t) + \epsilon_{\boldsymbol{a}}(t) < \widehat{\mu}_{\boldsymbol{b}}(t) - \epsilon_{\boldsymbol{b}}(t),$$

then with high probability, arm $\boldsymbol{a}$ is suboptimal and can be safely eliminated.

Motivated by this principle, we propose a new coordination strategy. Each player maintains a *desired set* of arms that could still potentially be optimal. As in Problem A, we fix a total ordering on the set of all joint arms:

$$\boldsymbol{a}_1 \xrightarrow{\hspace{1cm}} \boldsymbol{a}_2 \xrightarrow{\hspace{1cm}} \cdots \xrightarrow{\hspace{1cm}} \boldsymbol{a}_{K^{Md}} \tag{6}$$

At each round, players iterate through the joint arms that are still in the desired set, according to the fixed ordering. Suppose the next arm in the desired set is $\boldsymbol{a}$. If a player $i$ observes (based on their private reward samples) that some other arm $\boldsymbol{b}$ dominates $\boldsymbol{a}$—i.e., $I_{\boldsymbol{b}}(t)$ lies strictly above $I_{\boldsymbol{a}}(t)$—then player $i$ will intentionally deviate from $\boldsymbol{a}[i]$ and choose a different action.

Because actions are observable, other players will recognize that $\boldsymbol{a}$ was the next arm in line according to the fixed ordering, but that the actual joint action played differs from $\boldsymbol{a}$. This signals that some player has chosen to eliminate $\boldsymbol{a}$ from the desired set. As a result, all players will simultaneously remove $\boldsymbol{a}$ from their own desired sets. The total ordering ensures all players remain synchronized in the elimination process, without any need for explicit communication.

The full pseudocode is presented in Algorithm 2.

---

**Algorithm 2:** `mCAB-B` for asymmetry in actions

---
**1 Input:** $T, m, d \in \mathbb{N}$
**2** Each player will discretize their actions space as $\{0, \frac{1}{K}, \frac{2}{K}, ..., 1\}^d$, where $K$ is given in equation 4
**3** Run `mUCB-Intervals` on discretized joint action space Chang and Lu (2023)
**4** Each player $P_i$ has all the joint arms in their *desired sets*. All the players will agree on the ordering of the joint arms.
**5 for** $t = 1, \ldots, (K^d)^M$ **do**
**6** $\quad$ Each player $i$ will pull each joint action $\boldsymbol{a}$ once in the order they have decided in advance, and update $I_{\boldsymbol{a}}^i$.
**7 end**
**8 for** $t = K^M + 1, \ldots, T$ **do**
**9** $\quad$ Each player $i$ identifies the next arm considered $\boldsymbol{c}_t$ in the desired set based on the arm pulled in the previous round (see flowchart in equation 6).
**10** $\quad$ **if** *exists player $i$ and joint action $\boldsymbol{a}'$ such that $I_{\boldsymbol{a}'}^i$ is above and disjoint from $I_{\boldsymbol{c}_t}^i$* **then**
**11** $\quad\quad$ Player $i$ will not pull $\boldsymbol{c}_t[i]$ to inform the other players that he will remove $\boldsymbol{c}_t$ from his desired set.
**12** $\quad$ **else**
**13** $\quad\quad$ Each player $i$ pull $\boldsymbol{c}_t[i]$.
**14** $\quad$ **end**
**15** $\quad$ Each player $i$ observes the actions of other players to determine the joint action taken $\boldsymbol{a}_t$ at that step. They observe their own i.i.d. reward and update their $I_{\boldsymbol{a}_t}^i$.
**16** $\quad$ **if** $\boldsymbol{a}_t \neq \boldsymbol{c}_t$ **then**
**17** $\quad\quad$ All players eliminate $\boldsymbol{c}_t$ from their desired set whilst maintaining the same ordering of the remaining arms.
**18** $\quad$ **end**
**19 end**

---

We can now present the regret bound for the Problem B algorithm.

**Theorem 4** *The regret of* `mCAB-B` *is given by*

$$R_T = O\left(T^{\frac{2Md+1}{2Md+2}} L^{\frac{Md}{Md+1}} (\log T)^{\frac{1}{2(Md+1)}}\right)$$

**Proof:** In accordance to Theorem 2 Chang and Lu (2023) we know that the regret on `mUCB-Intervals` is the same as that in `mUCB` of Chang et al. (2022). Therefore using Lemma 1, we obtain the same regret bound as in Theorem 3. □

### 3.4 Asymmetry in both arms and rewards (Problem C)

## 4 Problem C: Asymmetry in Actions and Rewards

In this section, we propose an algorithm for **Problem C**, which models the multiplayer bandit setting with both *asymmetry in actions* and *asymmetry in rewards*. In this setting, each player neither observes the actions of the other players nor shares a common reward signal. Consequently, the algorithms designed for Problems A and B are no longer applicable: the algorithm for Problem A relies on synchronized reward observations, and the algorithm for Problem B assumes observability of actions.

To address this, we adapt the `mDSEE` algorithm from Chang et al. (2022) to the Lipschitz setting. This algorithm can be viewed as an improved *explore-then-commit* strategy. As before, the action space is uniformly discretized, and all players agree on a fixed ordering of the joint actions. The algorithm proceeds in phases, indexed by $n$, where phase $n$ begins at round $t = 2^n$.

Each phase consists of two stages:

- **Exploration phase:** Each player explores all $K^{md}$ joint arms uniformly, sampling each joint arm $f(n)$ times. This phase enables the players to accumulate statistically significant estimates of the reward distribution for each joint arm, despite having different samples.

- **Committing phase:** After exploration, each player independently computes empirical means for each joint arm based on all samples collected so far (from all earlier exploration phases). Then, each player commits to the joint arm with the highest empirical mean, using their own private samples. This committing phase lasts until the beginning of the next power-of-two round.

Because the number of rounds between exploration phases grows exponentially, the total number of samples used for exploration is bounded by a slowly growing function $f(\log T)$. This ensures that the regret from the exploration phases remains small, e.g., $O(f(\log T))$.

Moreover, during the committing phases, the players are increasingly likely to coordinate on the optimal joint action. This is because their empirical estimates become more accurate as the number of samples increases across phases. Over time, this leads to increasingly consistent coordination on the optimal joint action, despite the absence of shared feedback or observability.

The full pseudocode is provided in Algorithm 9.

Using the results from Chang et al. (2022), we note that on $K$ actions the regret of `mDSEE` is given by $O(f(\lfloor \log T \rfloor) \log T)$ where $f : \mathbb{N} \to \mathbb{N}$ is some function that satisfies $\lim_{n \to \infty} f(n) = \infty$.

**Theorem 5** *Using* $f(n) = \sqrt{n}$, *The (gap independent) regret bound for 9 is given by*

$$R_T \leq O\left(T^{\frac{2Md+1}{2Md+2}} L^{\frac{Md}{Md+1}} (\log T)^{\frac{1}{(Md+1)}}\right)$$

**Proof:** Suppose each player discretized their action space $[0, 1]$ into $K$ equal partitions. Using Theorem 2 of Chang et al. (2022) we know that the regret $R_K(T)$ on $K^{Md}$ joint actions is given by

$$R_T \leq \frac{TL}{K^{Md}} + O(K^{Md} \log T \sqrt{T})$$

---
**Algorithm 3:** `mCAB-C` for asymmetry in actions

---
**1** **Input:** $T, m, d \in \mathbb{N}$

**2** Each player will discretize their actions space as $\{0, \frac{1}{K}, \frac{2}{K}, ..., 1\}^d$, where $K$ is given by equation 7

**3** Run `mDSEE` on discretized joint action space Chang et al. (2022)

**4** Pick a monotonic function $f(n) : \mathbb{N} \to \mathbb{N}$ such that $\lim_{n \to \infty} K(n) = \infty$.

**5** First let $\lambda = 1$

**6** Player $P_i$ will start from his arm 1 and successively pull each arm $f(n)K_{i+1} \cdots K_M$ before moving to the next arm. He will repeat this entire epoch $K_1 \cdots K_{i-1}$ times.

**7** Player $P_i$ will calculate the sample mean $\mu_j^i(t)$ of the rewards they see for each $M$-tuple $a$ of arms.

**8** Player $P_i$ will choose the $M$-tuple arm with the highest sample mean and commit to his corresponding arm for up until the next power of 2. In case of a tie, pick randomly.

**9** When $T = 2^n$ for some $n \geq \lfloor \log_2(f(1)K_1 \cdots K_M) \rfloor + 1$, Player $P_i$ with repeat steps (5) - (8), incrementing $n$ by 1.

---

Notice the increased power of $\log T$. Setting the two terms equal to each other to mimize the above, we can set,

$$K = \frac{T^{\frac{1}{2(Md+1)}} L^{\frac{1}{M+1}}}{(\log T)^{\frac{1}{(Md+1)}}} \tag{7}$$

Therefore, our Lipschitz regret is given by

$$R_T = O\left(T^{\frac{2Md+1}{2Md+2}} L^{\frac{Md}{Md+1}} (\log T)^{\frac{1}{(Md+1)}}\right)$$

$\square$

## 5 A more adaptive zooming algorithm

In this section, we present a multiplayer extension of the *zooming algorithm* introduced by Kleinberg et al. (2008), tailored to address the information asymmetry settings in **Problems A** and **B**. Unlike the fixed discretization approaches described in earlier sections, the zooming algorithm adaptively refines the discretization of the action space. It focuses sampling effort on regions where the optimal joint arm is more likely to lie, thereby achieving improved empirical and theoretical performance.

This adaptive refinement allows for more efficient exploration in high-reward regions, especially in large or continuous action spaces. In the multiplayer setting, we extend the zooming framework to incorporate coordination under information asymmetry: in Problem A, where rewards are shared but actions are private, and in Problem B, where actions are public but rewards are private.

### 5.1 Problem A: Information Asymmetry in Actions

In this section, we propose a multiplayer zooming algorithm tailored to **Problem A**, where players face *asymmetry in actions*. A key challenge in adapting the single-player zooming algorithm to this setting arises during the *activation step*, when a new arm is introduced into the active set. In the classical single-player zooming algorithm Kleinberg et al. (2008), each time an uncovered region is discovered, a new arm is added by selecting a point from that region. However, in the multiplayer setting with a continuous joint action space, this becomes problematic.

The uncovered region is defined as the complement of a union of confidence balls centered at previously activated arms. This complement is generally non-convex and can exhibit highly irregular geometry, making it unclear how players—who cannot observe each other's actions—can agree on which arm to activate next. Without a shared view of the geometry, coordination is difficult.

To address this, we observe that all players maintain confidence balls of the same size (since rewards are shared and UCB indices evolve identically). This allows for implicit coordination by introducing a structured discretization scheme based on a *doubling grid*. We formalize this below.

**Definition 6 (Doubling Discretization)** *Let $[0,1]^m$ denote the unit cube. A discretization of the cube at level $n$ is given by the grid*

$$\left\{ \frac{i_1}{2^n}, \frac{i_2}{2^n}, \ldots, \frac{i_m}{2^n} \right\} \quad \text{for all } (i_1, \ldots, i_m) \in \{0, \ldots, 2^n\}^m.$$

*We say we* double the grid *if we increase the discretization level from $n$ to $n + 1$, i.e., the grid becomes $\left\{ \frac{i}{2^{n+1}} \right\}^m$. This results in $2^m$ times more cubes, each with $1/2^m$ the volume of the previous cubes.*

At any given time $t$, we define the set of *active arms* to be the subset of grid points currently under consideration. Each active arm $\boldsymbol{a}$ has an associated confidence ball centered at $\boldsymbol{a}$ with radius

$$r_{\boldsymbol{a}}(t) := \frac{1}{L} \cdot \epsilon_{\boldsymbol{a}}(t), \quad \text{where } \epsilon_{\boldsymbol{a}}(t) = \sqrt{\frac{6 \log T}{n_{\boldsymbol{a}}(t)}}.$$

The zooming algorithm proceeds by maintaining coverage of the action space via these confidence balls. When a region is no longer covered, the players implicitly agree to activate the next candidate arm from the current grid level (based on a fixed ordering). If all regions are covered, but the total number of active arms has not yet saturated the space, the players double the grid as described in Definition 6. This ensures implicit synchronization without requiring communication or observability of each other's actions.

---

**Algorithm 4: `mZoom-A`**

---

**1 Input:** $T, M, d \in \mathbb{N}$

**2** Each player will activate $(\frac{1}{2}, ..., \frac{1}{2}) \in A$ where $A$ is the active set

**3 for** $t = 1, ..., T$ **do**

**4**      If there is a region that is uncovered by the balls with radius $r_{\boldsymbol{a}}(t) = L\epsilon_{\boldsymbol{a}(t)}$ we will double the region (as in definition 6) until at least 1 point in our new discretization is in an unoccupied region. Add the arm with the smallest order relation as specified in Definition 2 into $\mathcal{A}$.

**5**      Player $P_i$ chooses arm $a_i^*(t) = \arg\max_{a \in \mathcal{A}} \left( \max_{a_{-i}} \text{UCB}_{a_i, a_{-i}}^i(t) \right)$ which corresponds to player $i$ picking the $i$th component of $a^*(t)$ that maximizes the index $\text{UCB}_a(t)$. In case of a tie between say $a$ and $a'$, they pick corresponding components of $a$ such that $a < a'$, where the order relation is as specified in Definition 2.

**6**      Player $P_i$ updates the UCB index $\text{UCB}_a^i(t+1)$ for arm $a$ setting $\delta = \frac{1}{T^2}$ with the received reward $X_{a^*(t)}^i(t)$.

**7 end**

---

The region is covered with balls that hold arms. If an uncovered region contains an arm, then a new ball is placed to cover the uncovered arm. As the regions are a discretized space, open arms are found via this discretization. If there are no open regions in the current discretized space, the space is discretized exponentially by 2. The discretized space is divided in half until an uncovered region is found. If there are multiple uncovered arms, the players can define an order to cover the arms, as there is a finite number of arms to choose from. The players will infer which arms the players will pick (including the newly created arms), explained in Definition 6.

The following gives us a regret bound for the regret of Algorithm 7

**Theorem 7** *The regret of `mZoom-A` in Algorithm 7 is*

$$R(T) = O\left( T^{\frac{Md+1}{Md+2}} (c \log T)^{\frac{1}{Md+2}} \right)$$

The proof is in section B of the appendix.

### 5.2 Problem B: Information Asymmetry in Rewards

In this section, we propose an adaptive algorithm for **Problem B**, the setting of information asymmetry in rewards. Our approach builds on the structure of Algorithm 2, incorporating adaptive discretization via the zooming technique described in the previous section.

As before, we cover the action space using confidence balls. Specifically, for each *active* joint arm $\boldsymbol{a}$, we place a ball of radius

$$r_{\boldsymbol{a}}(t) := \frac{1}{L} \cdot \epsilon_{\boldsymbol{a}}(t), \quad \text{where } \epsilon_{\boldsymbol{a}}(t) = \sqrt{\frac{6 \log T}{n_{\boldsymbol{a}}(t)}}.$$

New arms are activated whenever a region becomes uncovered by the current set of balls. Since the space of uncovered points is infinite and unstructured, we resolve this ambiguity by applying the same *doubling discretization* scheme as described in Definition 6. This ensures that all players have a shared understanding of which arms are eligible for activation.

The algorithm maintains the elimination-based structure of Algorithm 2. At any time, players iterate through the currently active arms according to a predefined total ordering (see Definition 2). The elimination rule remains unchanged: if the next joint arm in the ordering is $\boldsymbol{a}$, and a player $i$ observes another joint arm $\boldsymbol{b}$ such that the confidence interval $I_{\boldsymbol{b}}(t)$ lies strictly above and disjoint from $I_{\boldsymbol{a}}(t)$, then player $i$ intentionally deviates by selecting an action different from $\boldsymbol{a}[i]$.

Because actions are observable, the other players recognize that $\boldsymbol{a}$ was skipped, and therefore eliminate it from their own desired sets. This coordination mechanism enables players to maintain consistent elimination logic without requiring direct communication.

The full pseudocode is presented in Algorithm 5.

---

**Algorithm 5:** `Zoom-B` for asymmetry in actions

**1 Input:** $T, m, d \in \mathbb{N}$

**2** Each player will activate $(\frac{1}{2}, ..., \frac{1}{2}) \in A$ where $A$ is the active set, and add it to the desired set as well.

**3 for** $t = 1, ..., T$ **do**

**4**     If there is an arm that is uncovered, double the region until at least 1 point in the discretization is in an unoccupied region. Add the arm with the smallest order relation as specified in Definition 2 as the last element in the desired set.

**5**     Pull the new arm the same number of times as all the other arms in the desired set.

**6**     If there is another uncovered region, repeat steps 4 and 5 until the entire region is covered.

**7**     Each player $i$ identifies the next arm considered $\boldsymbol{c}_t$ in the desired set based on the arm pulled in the previous round.

**8**     **if** *exists player $i$ and joint action $\boldsymbol{a}, \in \mathcal{A}_t$ such that $\widehat{\mu}_{\boldsymbol{c}} + 2\epsilon_{\boldsymbol{c}} < \widehat{\mu}_{\boldsymbol{a}} - \epsilon_{\boldsymbol{a}}$* **then**

**9**        Player $i$ will not pull $\boldsymbol{c}[i]$ to inform the other players that he will remove $\boldsymbol{a}$ from his desired set.

**10**     **else**

**11**        Each player $i$ pull $\boldsymbol{c}_t[i]$.

**12**     **end**

**13**     Each player $i$ observes the actions of other players to determine the joint action taken $\boldsymbol{a}_t$ at that step. They observe their own i.i.d. reward and update $\widehat{\mu}_{\boldsymbol{a}_t}, \epsilon_{\boldsymbol{a}_t}$.

**14**     **if** $\boldsymbol{a}_t \neq \boldsymbol{c}_t$ **then**

**15**        All players eliminate $\boldsymbol{c}_t$ from their desired set whilst maintaining the same ordering of the remaining arms. The radius of the ball centered at this arm would then remain unchanged for the remainder of the learning.

**16**     **end**

**17 end**

---

Note that step 6 must terminate after a finite number of steps since every ball in the desired set has the same radius (meaning the radii cannot be arbitrarily small).

This algorithm is interesting since during the process new active arms are created while at the same time, arms already active are also being eliminated. Note that once an active arm $\boldsymbol{a}$ is eliminated, this also eliminates the arms distance $r_{\boldsymbol{a}}(t)$ from $\boldsymbol{a}$. Here $t$ is the round that $\boldsymbol{a}$ was eliminated. This is made explicit in step 8 of Algorithm 5, when the players eliminate an arm $\boldsymbol{a}$ from the desired set, they are in fact eliminating the ball of arms centered at $\boldsymbol{a}$ with radius $r_{\boldsymbol{a}}(t)$. We have to ensure that the ball that gets eliminated doesn't contain the optimal arm.

**Lemma 8** *The optimal arm, under the good event $G$, never gets eliminated.*

**Proof:** *Suppose that $\boldsymbol{c}$ was the arm that's getting eliminated that is, and and suppose that $\boldsymbol{a}$ is such that*

$$\widehat{\mu}_{\boldsymbol{c}} + 2\epsilon_{\boldsymbol{c}} < \widehat{\mu}_{\boldsymbol{a}} - \epsilon_{\boldsymbol{a}}$$

*Then all the arms $\boldsymbol{b} \in B(\boldsymbol{c}, r_{\boldsymbol{c}})$ satisfy the following (under the good event $G$).*

$$\mu_{\boldsymbol{b}} \leq \mu_{\boldsymbol{c}} + L\|\boldsymbol{c} - \boldsymbol{b}\| \tag{8}$$
$$\leq \mu_{\boldsymbol{c}} + Lr_{\boldsymbol{c}} \tag{9}$$
$$= \mu_{\boldsymbol{c}} + L\frac{1}{L}\epsilon_{\boldsymbol{c}} \tag{10}$$
$$= \widehat{\mu}_{\boldsymbol{c}} + \epsilon_{\boldsymbol{c}} + L\frac{1}{L}\epsilon_{\boldsymbol{c}} \tag{11}$$
$$< \widehat{\mu}_{\boldsymbol{a}} - \epsilon_{\boldsymbol{a}} \tag{12}$$
$$= \mu_{\boldsymbol{a}} \tag{13}$$

*Therefore $\boldsymbol{b}$ cannot be the optimal arm for its mean is less than or equal to another arm.*

$\square$

Using the above lemma we can prove the following regret bound

**Theorem 9** *The regret of `mZoom-B` in Algorithm 5 is*

$$R(T) = O\left(T^{\frac{Md+1}{Md+2}}(c\log T)^{\frac{1}{Md+2}}\right)$$

Note here that the constant in this theorem is different than the one in Theorem 7. The difference is illustrated in equations equation 32 and equation 34. The proof is given in section C of the appendix.

## 6 Conclusions

In this paper, we propose the first algorithmic framework for the multiplayer information-asymmetric Lipschitz bandit setting. We introduce three distinct problems—Problem A, B, and C—corresponding to information asymmetry in actions, rewards, and both, respectively. To address these challenges, we leverage a combination of uniform discretization and recently developed multi-agent algorithms from Chang et al. (2022); Chang and Lu (2023) to handle each form of asymmetry. Furthermore, we adapt the zooming algorithm to accommodate asymmetry in actions (Problem A) and rewards (Problem B). For Problems A and B, we prove regret bounds that match the best-known rates in the single-player Lipschitz bandit setting. For Problem C, which involves both forms of asymmetry simultaneously, we provide an algorithm with a regret bound that is nearly optimal up to logarithmic factors.

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

## A   Important Lemmmas

We have the following Lemma that serves as a concentration inequality for subgaussian random variables from Lattimore and Szepesvári (2020).

**Lemma 10** *Assume that $X_i - \mu$ are independent, $\sigma$-subgaussian random variables. Then for any $\varepsilon \geq 0$,*

$$\mathbb{P}(\hat{\mu} \geq \mu + \varepsilon) \leq \exp\left(-\frac{n\varepsilon^2}{2\sigma^2}\right) \quad and \quad \mathbb{P}(\hat{\mu} \leq \mu - \varepsilon) \leq \exp\left(-\frac{n\varepsilon^2}{2\sigma^2}\right),$$

*where $\hat{\mu} = \frac{1}{n}\sum_{t=1}^{n} X_t$.*

## B   Regret analysis for `mZoom-A` for asymmetry in actions

Consider the good event $G$ for each arm $\boldsymbol{a}$ at time $t$ be defined as

$$G_{\boldsymbol{a}}(t) = \{|\hat{\mu}_{\boldsymbol{a}}(t) - \mu_{\boldsymbol{a}}| \leq \epsilon_{\boldsymbol{a}}(t)\}$$

Furthermore, let $\mathcal{A}_t$ be the set of activated arms at round $t$ (note that we have $\mathcal{A}_t \subset \mathcal{A}_{t+1}$) and let $G$ be the good event that $G_{\boldsymbol{a}}(t)$ holds across any round $t$ and any activated arm. That is,

$$G = \bigcap_{t=1}^{T} \bigcap_{\boldsymbol{a} \in \mathcal{A}_t} G_{\boldsymbol{a}}(t)$$

The following lemma gives us a bound for the probability that this good event holds.

**Lemma 11** *The probability that $G^c$ holds is upper bounded by $\frac{1}{T}$. That is,*

$$P(G^c) \leq \frac{1}{T}$$

**Proof:**

$$\epsilon_{\boldsymbol{a}} = \sqrt{\frac{6\log(T)}{n_a(t)}} \tag{14}$$

We first find an upper bound on the probability of the 'bad' event for each joint action $G_{\boldsymbol{a}}(t)$. Using lemma 10, we know we have the following,

$$P(\hat{\mu}_a \geq \mu_a + \epsilon_a(t)) \leq e^{-\frac{n_a(t)\left(\sqrt{\frac{6\log(T)}{n_a(t)}}\right)^2}{2}} \tag{15}$$

$$= e^{\frac{-n_a(t)\cdot 6\log(T)}{2n_a(t)}} \tag{16}$$

$$= e^{-3\log(T)} \tag{17}$$

$$= (e^{\log(T)})^{-3} \tag{18}$$

$$= T^{-3} \tag{19}$$

where we used the definition of $\epsilon_{\boldsymbol{a}}$ in the first inequality. Using Demorgans rule, we now have

$$G^c = \bigcup_{t=1}^{T} \bigcup_{\boldsymbol{a} \in \mathcal{A}_t} G_{\boldsymbol{a}}(t)^c$$

Therefore, by our probability union bound, we can upper bound the probability of the 'bad' event,

$$P(G^c) = P\left(\bigcup_{t=1}^{T} \bigcup_{\boldsymbol{a} \in \mathcal{A}_t} G_a^c(t)\right) \tag{20}$$

$$= \sum_{t=1}^{T} \sum_{\boldsymbol{a} \in \mathcal{A}_t} P(G_a^c(t)) \tag{21}$$

$$= \sum_{t=1}^{T} \sum_{\boldsymbol{a} \in \mathcal{A}_t} \frac{1}{T^3} \tag{22}$$

$$= \sum_{t=1}^{T} |\mathcal{A}_t| \frac{1}{T^3} \tag{23}$$

$$\leq \sum_{t=1}^{T} \frac{1}{T^2} \tag{24}$$

$$= \frac{1}{T} \tag{25}$$

where in the 3rd inequality we used our upper bound above for $P(G_{\boldsymbol{a}(t))})$. In the 4th equality we used the fact that $|\mathcal{A}_t| \leq T$ because we are only adding at most 1 arm per found to our active set. $\qquad\square$

We can now prove Theorem 7

**Proof:** Note that the regret decomposition by the Law of Total Expectation is,

$$R_T \leq \mathbb{E}[R_T|G]P(G) + \mathbb{E}[R_T|G^c]P(G^c)$$

Convince yourself that $\mathbb{E}[R_T|G^c] \leq T$ and $P(G) \leq 1$. You also proved already that $P(G^c) \leq \frac{1}{T}$. Therefore,

$$R_T \leq \mathbb{E}[R_T|G] + 1$$

We now calculate the regret assuming that the good event $G$ happens.

Since the algorithm pulls the arm with the highest regret every round, if $\boldsymbol{a}_t$ was played, this implies

$$\text{UCB}_{\boldsymbol{a}_t}(t) \geq \text{UCB}_{\boldsymbol{a}_t^*}$$

where $\boldsymbol{a}_t^*$ is arm in the active set $\mathcal{A}_t$ that covers the true optimal arm $\boldsymbol{a}^\star$. Now note

$$\text{UCB}_{\boldsymbol{a}_t} \geq \text{UCB}_{\boldsymbol{a}_t^*} \tag{26}$$

$$\geq \widehat{\mu}_{\boldsymbol{a}_t^*} + \epsilon_{\boldsymbol{a}_t}(t) + \epsilon_{\boldsymbol{a}_t}(t) \tag{27}$$

$$\geq \mu_{\boldsymbol{a}_t^*} + L\|\boldsymbol{a_t} - \boldsymbol{a}^\star\| \tag{28}$$

$$\geq \mu^* \tag{29}$$

Furthermore,

$$\text{UCB}_{\boldsymbol{a}_t}(t) = \widehat{\mu}_{\boldsymbol{a}_t}(t) + 2\epsilon_{\boldsymbol{a}_t}(t) \tag{30}$$

$$\leq \mu_{\boldsymbol{a}_t} + 3\epsilon_{\boldsymbol{a}_t}(t) \tag{31}$$

Combining the upper and lower bound for $\text{UCB}_{\boldsymbol{a}_t}$ above, we conclude that under the good event, for each arm $\boldsymbol{a} \in \mathcal{A}$, we have $\Delta_{\boldsymbol{a}} \leq 3\epsilon_{\boldsymbol{a}}(t)$ for each time $t$. Note that this is true regardless the arm is pulled at that time $t$ or not.

From the definition of $\epsilon_{\mathbf{a}_t}(t)$ in equation equation 14,

$$\Delta_{\mathbf{a}} \leq 3\sqrt{\frac{6 \log T}{n_{\mathbf{a}}(t)}}$$

Rearranging the above, we conclude that

$$n_{\mathbf{a}}(t) \leq \frac{54 \log T}{\Delta_{\mathbf{a}}^2} \tag{32}$$

We now proceed as in the proof of Theorem 5 but with eqution equation 32 replacing equation equation 34. $\square$

## C   Regret analysis for `mZoom-B` for asymmetry in rewards

**Proof:**   We define the following "good" event $G$ where all arms have their true means in the intervals at all times. Explicitly, this is written as

$$G = \bigcap_{t=1}^{T} \bigcap_{i=1}^{M} \bigcap_{\mathbf{a} \in \mathcal{A}} \{|\hat{\mu}_{\mathbf{a}}^i(t) - \mu_{\mathbf{a}}| < \epsilon_{\mathbf{a}}(t, \delta)\} \tag{33}$$

The probability of the complement of this event is already bounded in the proof of Theorem 11.

By lemma 8, the best arm is never in a region tha is eliminated. Therefore at round $t$, we can call $\mathbf{a}_t^*$ the arm whose ball contains the best arm. As all the arms are pulled in a round-robin fashion in the desired set, if an arm $a$ is at the desired set at time $t$ then $n_{\mathbf{a}^*}(t) \geq n_{\mathbf{a}}(t) - 1$. It follows that at the time when $n_{\mathbf{a}}(t) = n_{\mathbf{a}}$, we have $n_{\mathbf{a}^*}(t) \geq n_{\mathbf{a}} - 1$. When each arm in the desired set has been pulled at least twice, this means that $n_{\mathbf{a}^*}(t) \geq \frac{1}{2} n_{\mathbf{a}}$. Under the good event note that

$$\hat{\mu}_{\mathbf{a}}(t) \leq \mu_{\mathbf{a}} + \epsilon_{\mathbf{a}}(t) \implies \mathrm{UCB}_{\mathbf{a}}(t) \leq \mu_{\mathbf{a}} + 2\epsilon_{\mathbf{a}}(t) = \mu_{\mathbf{a}} + 2\sqrt{\frac{6 \log T}{n_{\mathbf{a}}(t)}}$$

We we recalled the definition of $\epsilon_{\mathbf{a}}(t)$ give in equation equation 14 Similarly,

$$\mathrm{LCB}_{\mathbf{a}_t^*} \geq \mu_{\mathbf{a}_t^*} - 2\epsilon_{\mathbf{a}_t^*}(t) \geq \mu_{\mathbf{a}_t^*} - 2\sqrt{\frac{6 \log T}{n_{\mathbf{a}}(t)/2}}$$

Furthermore, note that

$$\mu_{\mathbf{a}^\star} \leq \mu_{\mathbf{a}_t^*} + Lr_{\mathbf{a}}(t) \leq \mu_{\mathbf{a}_t^*} + \epsilon_{\mathbf{a}_t^*}(t) \leq \mu_{\mathbf{a}_t^*} + \sqrt{\frac{6 \log T}{n_{\mathbf{a}}(t)/2}}$$

Therefore,

$$\mathrm{LCB}_{\mathbf{a}_t^*}(t) \geq \mu_{\mathbf{a}^\star} - 3\sqrt{\frac{6 \log T}{n_{\mathbf{a}}(t)/2}}$$

It follows that if $\mathbf{a}_t$ is pulled, it is in the desired set so that $\mathrm{LCB}_{\mathbf{a}_t^*}(t) \geq \mathrm{UCB}_{\mathbf{a}}(t)$, it follows that we have the following necessary condition

$$\mu_{\mathbf{a}^\star} - 3\sqrt{\frac{6 \log T}{n_{\mathbf{a}}(t)/2}} \leq \mu_{\mathbf{a}} + 2\sqrt{\frac{6 \log T}{n_{\mathbf{a}}(t)}}$$

Rearranging the above gives

$$n_{\mathbf{a}}(t) \leq (6\sqrt{3} + 2\sqrt{6})^2 \frac{\log T}{\Delta_{\mathbf{a}}^2} \tag{34}$$

For $r > 0$, consider the set of active arms whose badness is between $r$ and $2r$ :

$$X_r = \{x \in \mathcal{A}_t : r \leq \Delta_{\boldsymbol{a}} < 2r\}.$$

Fix $i \in \mathbb{N}$ and let $Y_i = X_r$, where $r = 2^{-i}$. From the bulletpoint above, for any $\boldsymbol{a}, \boldsymbol{a}' \in Y_i$, we have $d(\boldsymbol{a}, \boldsymbol{a}') > \frac{1}{3L}\Delta_{\boldsymbol{a}}$. If we cover $Y_i$ with subsets of diameter $\frac{r}{3L}$, then arms $\boldsymbol{a}$ and $\boldsymbol{a}'$ cannot lie in the same subset. Since one can cover $Y_i$ with $N_{\frac{r}{3L}}(Y_i)$ such subsets, it follows that $|Y_i| \leq N_{\frac{r}{3L}}(Y_i)$.

Note that in the following analysis we don't really care of $\Delta_{\boldsymbol{a}} = 0$ since we will split the good arms from the bad, the following upper bound will only be used on the bad arms. From equation equation 34, we have:

$$R_i(T) := \sum_{\boldsymbol{a} \in Y_i} \Delta_{\boldsymbol{a}}\mathbb{E}[n_{\boldsymbol{a}}(T)] \leq \sum_{\boldsymbol{a} \in Y_i} \Delta_{\boldsymbol{a}} \cdot \frac{O(\log T)}{\Delta_{\boldsymbol{a}}^2} \leq \frac{O(\log T)}{\Delta_{\boldsymbol{a}}} \cdot N_{r/3}(Y_i) \leq \frac{O(\log T)}{r} \cdot N_{r/3}(Y_i).$$

Pick $\delta > 0$, and consider arms with $\Delta(\cdot) \leq \delta$ separately from those with $\Delta(\cdot) > \delta$. Note that the total regret from the former cannot exceed $\delta$ per round. Therefore:

$$R(T) \leq \delta T + \sum_{i:r=2^{-i}>\delta} R_i(T)$$

$$\leq \delta T + \sum_{i:r=2^{-i}>\delta} \frac{\Theta(\log T)}{r} N_{r/3}(Y_i)$$

$$\leq \delta T + O(c \cdot \log T) \cdot \left(\frac{1}{\delta}\right)^{d+1},$$

where $c$ is a constant and $d$ is some number such that

$$N_{r/3}(X_r) \leq \frac{c}{r^d} \quad \forall r > 0.$$

the last inequality then follows from $r = 2^{-i} > \delta$.

The smallest (infimum) such $d$ is called the zooming dimension with multiplier $c$. By choosing $\delta = \left(\frac{\log T}{T}\right)^{1/(Md+2)}$, we obtain

$$R(T) = O\left(T^{\frac{Md+1}{Md+2}}(c\log T)^{\frac{1}{Md+2}}\right).$$

Note that we make this choice in the analysis only; the algorithm does not depend on the $\delta$.

$\square$

