# OpenReview forum: "Multiplayer Information Asymmetric Bandits in Metric Spaces"
_TMLR — Rejected by TMLR_

### Review · Reviewer_h8q7 · 2025-08-16

**Summary Of Contributions:**

This paper considers the problem of multi-armed bandits in a setting that is multi-agent, with discrete actions but Lipschitz continuity, and partial observability of information. The problem considers three settings: one with limited visibility of actions (the chosen arms), one where agents receive different samples of rewards from the same underlying distributions, and one with both aspects. The authors apply discretization, which then allows for the use of discrete multi-armed bandit algorithms. The three settings are treated in succession and their regret bounds are determined. Lastly, an adaptive discretization approach is considered which is not uniform.

**Audience:**

Yes

**Audience Explanation:**

Yes, this paper falls well within the remit of TMLR, and studies a variant of a classic problem in reinforcement learning.

**Claims And Evidence:**

No

**Claims Explanation:**

See M1/M2/M3 in my review below.

**Requested Changes:**

I would like to preface my comments by noting that I am not an expert in multi-armed bandits and my review is fairly high-level as a result. While I do understand the key topic and results, I am unable to comment on the correctness in detail. I am also uncertain whether the level of contribution is appropriate.

### Major Comments

M1. At a high level, the paper takes algorithms developed for discrete problems with similar assumptions and adapts them to the Lipschitz setting. The core challenges treated by this paper are, in fact, to develop  coordination mechanisms so that previous learning algorithms can apply. In my opinion, the coordination algorithms should be described formally and their correctness should be proved. This is an area the paper should focus more on in my opinion. The adaptation of prior results appears relatively straightforward in comparison.

M2. In particular, I am uncertain about the solution for Problem B (asymmetry in rewards). Given players are working with finite samples, it may be the case that a player decides to wrongly eliminate an arm whose bounds appear worse than its values in expectation. This error will propagate to the other players. Is the regret bound still applicable in this case? Can something be said about how likely such errors are?

M3. I realise that this work is mainly theoretical, but I would expect to see experimental validation at least on toy problems. This would better motivate the study of this setting. As someone who is more empirical, the setting appears convoluted.

### Smaller Comments

S1. The abstract clearly needs editing (first few sentences look like an error).

S2. Section 2, $\mathbf{a}t$ should be $\mathbf{a}_t$ in several places.

S3. Algorithm 2, 3, 5 captions erroneously state algorithms are for asymmetry in actions only

---

### Review · Reviewer_UkCG · 2025-08-26

**Summary Of Contributions:**

This work introduces a framework for multiplayer Lipschitz bandits with information asymmetry, where agents face partial observability of rewards or actions. Three problem variants are studied: shared rewards but hidden actions (A), observed actions but private rewards (B), and both hidden actions and private rewards (C). The authors propose algorithms combining fixed discretization (mCAB) and adaptive zooming (mZoom) with multi-agent coordination mechanisms, achieving near-optimal regret bounds.

**Strengths:**

1. The work introduces deterministic arm orderings for implicit synchronization and elimination-based protocols to handle private feedback.
2. The methodology is well-described, with detailed proofs and pseudocode provided for reproducibility.


**Weaknesses:**

1. This paper does not provide a complete Abstract.
2. The introduction lacks a clear connection between the single-player case, multiplayer bandit settings, and information asymmetry. A stronger narrative linking these concepts would improve readability.
3. The section partitioning is unclear, particularly in Section 3.4 and Section 4.
4. The paper does not include experiments to verify the theoretical findings, limiting practical insights.

**Audience:**

Yes

**Audience Explanation:**

the paper addresses an intersection between multiplayer bandits and Lipschitz continuity that will interest researchers in bandit theory and multi-agent systems.

**Broader Impact Concerns:**

No ethical concerns are noted.

**Claims And Evidence:**

Yes

**Claims Explanation:**

This paper's theoretical claims are well-supported through rigorous proofs. However, the lack of empirical validation through experiments weakens the assessment of practical performance and scalability.

**Requested Changes:**

The abstract should be refined for conciseness and clarity. The paper needs careful reorganization to improve logical flow. The authors should consider adding experiments to validate the theoretical results. Additionally, the introduction would benefit from a stronger motivation that clearly connects the single-player case to multiplayer settings with information asymmetry.

---

### Review · Reviewer_UCFP · 2025-09-09

**Summary Of Contributions:**

The paper considered multiplayer (or say multiagent) bandits with the Lipschitz setting. It proposed a bunch of algorithms based on the framework of mCAB and mZoom for three multiplayer settings and provided upper bounds on the regret of algorithms.

**Audience:**

No

**Audience Explanation:**

I failed to find motivation of this work, comparion to existing algorithms, discussion on derived bounds, lower bounds or numerical experiments. Besides, most proofs are with little detail. Hence, it is difficult to understand the contribution of this work.

**Claims And Evidence:**

No

**Claims Explanation:**

1. The proofs need to be improved. For example, Proof of Theorem 3 claimed that '$R_T\le \frac{ L\sqrt{Md}\cdot T }{K} + O(\ldots)$'. The big O notation is used unproperly.
2. Many proofs are skipped and claimed to follow the analysis in existing literature, and hence it is hard to check whether the analysis is fine.

**Requested Changes:**

The author(s) should try to highlight the contributions (some suggestions are as above) and also improve the writing. For example,

1. the abstract begins with 'In recent years the information asymmetric Lipschitz bandits In this paper we studied the Lipschitz bandit problem applied to the multiplayer information asymmetric'. It is strange to begin the abstract with an incomplete sentence.
2. After equation (1): ' Intuitively, this condition implies that if an agent has a good estimate of the mean reward at one action, then nearby actions must have similar rewards. '
     - To my understanding, the similarity among rewards is not affected by the quality of estimate of mean rewards.

In short, the whole manuscript needs a carefull revision.

---

### Decision · Action_Editor_GfRF · 2025-10-14

**Recommendation:** Reject

**Audience:**

Yes

**Audience Explanation:**

Most the reviewers believe the topic of the paper falls within the scope of TMLR. Unfortunately, the evidence presented to support the claims is incomplete.

**Claims And Evidence:**

No

**Claims Explanation:**

The reviewers raise several concerns regarding the incompleteness of the evidence supporting the claims in the paper which have gone unanswered by the authors. I therefore believe the paper is not yet ready for publication.

From EIC - here are more details on the gap between the claims and the evidence, based on further reviewer discussions:

_More concretely, the bounds given in the paper are derived directly from discretizing the actions, thus inheriting them from previous single-agent algorithms. The multi-agent coordination components of the proposed algorithms are described ambiguously and not proven to be correct; and these components are arguably a much bigger departure from the original single-agent MAB algorithms. The proposed methods are also not evaluated empirically, so there is also no evidence that, despite vagueness in their description and lack of guarantees, they nevertheless work in practice. Since the authors did not respond to the reviews, these key aspects have remained unaddressed._